# Analysis of the Expansion Characteristics and Carrying Capacity of Oasis Farmland in Northwestern China in Recent 40 Years

Bofei Li [1,2], Dongwei Gui [2,3,*], Dongping Xue [2,3], Yunfei Liu [2,3], Zeeshan Ahmed [2,3] and Jiaqiang Lei [2,3]

1 College of Ecology and Environment, Xinjiang University, Urumqi 830017, China
2 State Key Laboratory of Desert and Oasis Ecology, Xinjiang Institute of Ecology and Geography, Chinese Academy of Sciences, Urumqi 830011, China
3 Cele National Station of Observation & Research for Desert Grassland Ecosystem in Xinjiang, Cele 848300, China
* Correspondence: guidwei@ms.xjb.ac.cn; Tel.: +86-0991-7885507

**Abstract:** An oasis is a unique landscape that fuels human subsistence and socioeconomic development in the desert ecosystem. However, the overexpansion of oases, especially farmlands, poses severe threats to available land and water resources. This study aims to assess the expansion levels, carrying capacity dynamics, and planting structure optimization to maximize economic returns in northwest China's five Typical Oasis Groups (5TOGs) using uniform data sources, time scales, and methods. Satellite products and a water-heat balance model were used to evaluate the changes in the area and carrying capacity dynamics of the 5TOGs. A linear programming approach was used to optimize each oasis's cropping structure for the carrying level scenario. The results showed that the area of 5TOGs has expanded from 1980–2020, and the increment of oasis farmland is the main driver of oasis expansion. The most dramatic expansion of oases and their farmlands occurred during 2010–2020. As a consequence, the carrying capacity of each oasis is deteriorating with this expansion. The additional water resources to support this expansion of the oases and their farmlands come from groundwater, which is declining rapidly. Based on the optimized planting structure, cotton remains the main crop in Xinjiang oases with more than 60% area, the cotton area should be reduced in the Hotan River Oases, and the planting structure of the Heihe River Oasis will remain unchanged. The findings of this study have provided a quantitative analysis of oasis expansion and planting structure optimization, which have practical implications for water resource management and sustainable development of agriculture to maintain the stability of the oasis ecosystem.

**Keywords:** carrying capacity; groundwater exploitation; oasis expansion; oasis farmland; planting structure; resource management

## 1. Introduction

Oasis, known as the "pearl of the sand sea," is a heterogeneous landscape developed from a stable water supply under the desert environment [1]. Oasis landscapes cover about 5% of China's total arid and semi-arid regions, supporting more than 90% of the population and 95% of socio-economic development [2,3]. A large proportion of oasis landscapes is usually occupied by agricultural farmlands [4,5]. Therefore, water utilization rates in the oasis landscape are almost 100%, and more than 90% of the available water is used for agricultural activities [6]. Hence, the sustainable development of the oasis relies heavily on the sustainable use of water resources, and agricultural water management is the key to sustainable water resource management in the oasis ecosystem.

The continuously increasing population and the corresponding rise in food demands drive the expansion of oases areas [7]. For instance, Xie et al. [8] reported a 60% expansion of oases in the Heihe River Basin from 1963 to 2013. Bie and Xie [9] revealed a linear increase in the area change of the oasis in the region, while Liu and Shen [10] demonstrated that the area change of the Heihe River Basin Oasis was non-linear, with a significant increase

during 2000–2010. Similarly, the oases in Xinjiang are also expanding, but the quantitative results given by various scholars about expansion are variable [1,11–13].

Due to the rapid oasis expansion, farmlands are now extending into the desert ecotones with large amounts of water consumption [14,15], thus putting enormous pressure on the oasis environment and creating an ecological crisis. Therefore, to ensure sustainable socio-economic development and environmental protection, an assessment of the carrying capacity of the oasis is inevitable.

Studies related to the carrying capacity of oases have mainly focused on water resource carrying capacity (WCC), ecological carrying capacity (ECC), and suitable scale (sustainability). For example, Meng et al. [16] studied the water resources exploration and carrying capacity of different regions of the Tarim Basin in China and concluded that the water resources in the region are highly exploited, and the remaining carrying capacity is minimal. Han and Jia [17] evaluated the water resource carrying capacity of Xinjiang, China, and argued that the population of Xinjiang was not overloaded. Similarly, Zhi et al. [18] assessed the urban water carrying capacity of Xinjiang's northern slopes of the Tianshan Mountains. They concluded that the urban water resources in the region were not overloaded.

In contrast, Wan et al. [19] concluded that water resources in the Tarim Basin, the Tuha Basin, the Hexi Corridor, and north of the Tianshan Mountains exhibit severe vulnerability. Wang et al. [20] demonstrated that ECC in the Xinjiang mountain-oasis-desert area increased considerably in the oasis region and decreased significantly in the southwestern desert region. The overall trend of ECC was decreasing. Ye et al., Hao et al., Guo et al., and Ling et al. [21–24] evaluated the suitable scales of the Ejinna Oasis, Hehe Oasis, Hotan Oasis, and Manas River Oasis, respectively. Although the above cases analyze the carrying capacity of oases from multiple perspectives, their methods, subjects, data sources, and time periods are different, which could lead to varying results in the same study area. Therefore, such inconsistent results cannot warrant a more holistic quantification and analysis of the oasis-carrying capacity.

Northwest China is located in the interior of Eurasia, with a typical arid environment having low precipitation and high evapotranspiration [25]. This region covers nearly 30% of China's land area, but its water resources account for only 5% of China's total water resources [26]. Due to the presence of China's major inland rivers in this region, such as the Tarim river, Heihe river and Shule river, the oases of the region have become important inland river irrigation areas in China for food, fiber and vegetable production [27–30]. At the same time, as a result of large-scale, artificially driven expansion, the oases of northwest China have formed oasis groups spanning multiple watersheds and are dominated by farmland on the landscape. Among them, the five typical oasis groups are Tianshan Mountain northern slope rivers oasis group (TMNO), the Aksu River Oasis Group (AKO), Hotan River Oasis Group (HTO), Yarkand River Oasis Group (YKO) and Heihe River Oasis Group (HHO). In this study, a holistic analysis of the 5TOGs is conducted using the unified method, time scale and data source.

This study aims to provide an integrated quantification of the expansion characteristics and changes in the carrying capacity of the 5TOGs and their oasis farmlands. The particular objectives of this study include: (1) To quantify the expansion characteristics of the 5TOGs and their oasis farmlands from 1980 to 2020 using a transfer matrix; (2) To estimate the change patterns of the carrying capacity of the 5TOGs by using a water-heat balance model and the GRACE products; (3) To calculate the suitable scales of the 5TOGs under current and future water resource conditions; (4) To provide a yield-maximizing crop planting pattern for achieving sustainable development of the oasis by using linear programming based on the above results. The findings of this study are expected to help in informed decision-making regarding the rational utilization of water resources and sustainable development of agriculture without compromising the stability of oases ecosystems.

## 2. Materials and Methods

### 2.1. Study Area

The 5TOGs in northwestern China were selected as the study area in this study. Among them, the HHO is located in Gansu Province, China, and belongs to the middle reaches of the Heihe river basin. HHO is comprised of many farmlands and has become the main runoff consumption area in the basin [31]. TMNO is located on the northern slope of the western section of the Tianshan Mountains in Xinjiang, China. This agricultural–industrial oasis complex involves several watersheds and irrigation areas, including the Urumqi, Shihezi, and Manas rivers, and the Changji, Manas, Shihezi, and Mosowan Irrigation Districts. AKO, HTO, and YKO are all located in the Tarim Basin in southern Xinjiang, China. Among them, AKO is located on the northwest edge of Tarim Basin, HTO is in the southwest of Tarim Basin, and YKO is in the west of Tarim Basin.

The 5TOGs are warm continental arid zones with limited water, sparse rain, intense evaporation and a large temperature difference among days and years [32]. The landscape presents a macroscopic pattern of wetland-forest (grass)–artificial farmland–desert vegetation along river corridors [33]. The geographical location of the five oasis groups is shown in Figure 1. Some basic information about the 5TOGs can be seen in Tables 1 and A1.

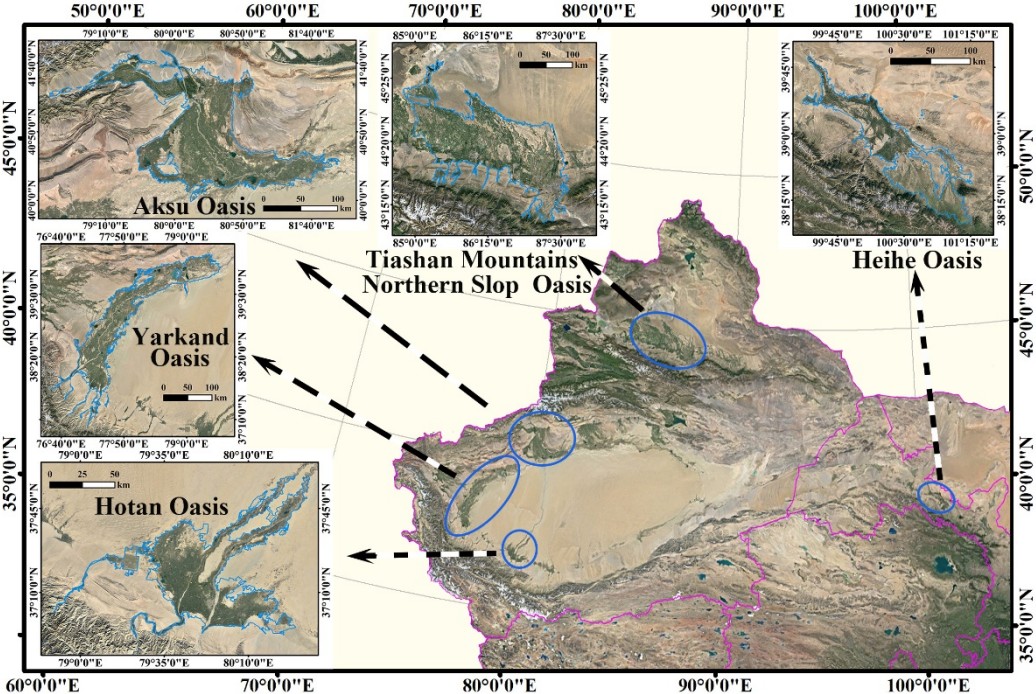

**Figure 1.** Location and landscape of the study area. The study area is located in northwest China as a whole. Among them, the Tianshan Mountains Northern Slop Oasis is in the north of Xinjiang, serving as the most important area for Xinjiang. The Aksu Oasis, Yarkand Oasis and Hotan Oasis are in Tarim Bain, south of Xinjiang. The Heihe Oasis is in the Hexi corridor, which contains several watersheds, connecting the mainland and western China since ancient times. Red lines: national boundaries and provincial boundaries; Blue circles: the locations of the oases; Blue lines: the Oasis boundaries after vectorization.

**Table 1.** Basic information of the five typical oasis groups.

| Items | Aksu River Oasis Group (AKO) | Hotan River Oasis Group (HTO) | Yarkand River Oasis Group (YKO) | Tianshan Mountain Northern Slope Rivers Oasis Group (TMNO) | Heihe River Oasis Group (HHO) |
|---|---|---|---|---|---|
| Area (km$^2$) | 12,293.65 | 4599.86 | 14,480.20 | 20,320.31 | 6836.77 |
| Farmland area (km$^2$) | 9241.69 | 3024.90 | 10,693.64 | 13,988.53 | 4302.85 |
| Available runoff volume ($\times 10^8$ m$^3$) | 53.42 | 30.27 | 64.75 | 39.00 | 20.93 |
| Urban and Industrial water use ($\times 10^8$ m$^3$) | 1.97 | 2.418 | 1.80 | 16.73 | 1.41 |
| Precipitation (mm) | 103.86 | 51.58 | 68.60 | 300.93 | 137.09 |
| Potential evapotranspiration (mm) | 1258.02 | 1363.14 | 1202.79 | 1110.72 | 1257.25 |

*2.2. Data Sources*

The area of the 5TOGs from 1980 to 2020 was calculated using the Google Earth platform; our team vectorized the boundaries of these oases and published the datasets through the Global Change Research Data Publishing and Repository (http://geodoi.ac.cn/WebEn/Default.aspx, accessed on 14 July 2020) [34–36]. The land use datasets were obtained from two platforms: the National Tibetan Plateau Data Center with a spatial resolution of 1 km and the Copernicus Open Access Hub with a spatial resolution of 10 m, by which the land use/cover data of the 5TOGs for 1980, 1990, 2000, 2010, and 2020 were extracted.

The meteorological observation data are collected from the China Meteorological Administration (http://data.cma.cn/, accessed on 14 July 2020), an official in China website for releasing quality meteorological data.

The ASTER GDEM V3 dataset [37] and the Google Earth platform were used to identify the actual area of the watersheds of 5TOGs. The annual average runoff data of each major oasis river was obtained from the Encyclopedia of Rivers and Lakes in China, published by China Water and Power Press [38].

Data regarding industry, population, and agriculture for the 5TOGs were obtained from the statistical yearbooks of the counties involved in these oases for the period 1980–2020. Irrigation reference volumes for the 5TOGs were obtained by checking the agricultural irrigation quota standards of the Northwest Region.

Data about groundwater storage change was extracted through the GRACE (the Gravity Recovery and Climate Experiment) RL06 Mascon Solutions (version 02) data with the spatial resolution of 0.25° × 0.25°. The data period was from April 2002 to August 2020 (cubic interpolation was used for some missing months). Data of terrestrial water storage (TWS) components were extracted from The Global Land Data Assimilation System (GLDAS) (available at https://disc.gsfc.nasa.gov/datasets?keywords=GLDAS, accessed on 21 September 2020). This part of the data has been validated by years of field groundwater monitoring data [39].

*2.3. Methods*

2.3.1. Water and Energy Balance Model for Oasis-Carrying Capacity

The oasis-carrying capacity ($H_0$) was estimated from a water-heat balance model [40] based on the match between water resources and heat exposure in an oasis region. It efficiently reflects the "stability" or "greenness" of the oasis under water stress. The closer the $H_0$ is to 1, the more stable the oasis is; when $H_0$ is less than 0.5, the oasis is considered unstable. The oasis-carrying capacity ($H_0$) is given by the following:

$$H_0 = \frac{W - W' + A \times r}{ET_0 \times A} \tag{1}$$

where $W$ is the total available water volume of the oasis ($10^8$ m$^3$), $W'$ is the total industrial and domestic water consumption ($10^8$ m$^3$), $A$ is the total area of the oasis (km$^2$), $r$ is the annual average precipitation (mm), and $ET_0$ is the reference crop evapotranspiration (mm), which is calculated by the Penman–Monteith formula [41]. The stability (Carrying Capacity) index of the oasis was classified in Table 2.

**Table 2.** Classification of oasis-carrying capacity (stability).

| $H_0$ | Type | Evaluation of Oasis Stability |
|---|---|---|
| >1.00 | Extremely stable | Oasis area has potential for expansion |
| 0.70–1.00 | Stable | Oasis area can be prudently expanded with reliable measures |
| 0.50–0.70 | Metastable | Oasis requires a high level of investment to remain stable |
| <0.50 | Unstable | Oasis must be reduced in size to maintain local stability |

According to Equation (1), the suitable oasis scale ($A'$) is given by the following:

$$A' = \frac{W - W'}{(ET_0 - r) \times H_0^*} \tag{2}$$

where $A'$ is the suitable scale of the oasis, which will serve as a foundation for optimizing the planting structure later. $H_0^*$ is set to 0.75 in this paper, which is the lower limit for the oasis-carrying capacity level in a stable state. The remaining parameters are the same as in Equation (1).

2.3.2. The Ground Water Storage Anomaly (GWSA) from GRACE and GLDAS

The changes in groundwater storage of 5TOGs were also quantified through GRACE and GLDAS to verify the changes in the carrying capacity of the oasis. The GRACE data reflect the terrestrial water storage ($TWS$) changes but cannot identify the source of the change. Similarly, the Noah-LSM model of GLDAS provides data on the changes in each terrestrial water component but does not reflect the overall change in terrestrial water storage. Therefore, by using GWSA, which integrates the GRACE and GLDAS models, changes in groundwater storage can be identified separately for rapid evaluation of oasis sustainability. It has been found that: (1) GWSA is a suitable and reliable indicator for trend change analysis [39], and (2) the Mann-Kendall (M-K) test results of long-term trend changes in GWSA are positively correlated with water resource carrying capacity (WRCC) [39].

$$\Delta TWS = \Delta GWS + \Delta SWE + \Delta SM \tag{3}$$

where $\Delta TWS$ (TWSA) consists of the $GWS$, $SWE$ and $SM$ anomalies of the Noah–LSM model of GLDAS, and all variables have been converted into equivalent water height so that distance units can represent them. Hence, $\Delta TWS$ can be extracted by the following:

$$\Delta GWS = \Delta TWS - \Delta SWE - \Delta SM \tag{4}$$

The changes in groundwater storage ($\Delta GWS$) are presented as equivalent water height (EWH mm/month).

2.3.3. Linear Programming for Planting Structure Optimization

After obtaining information on the suitable scales, farmland area data and planting structure of the 5TOGs, we optimized the planting structure of these oases to maximize economic benefits. The specific linear programming function is as follows:

$$max f = \sum_{i=1}^{I} a_i' w_i e_i p_i \tag{5}$$

Set of constraints:

$$
\begin{cases}
\sum\limits_{i=1}^{I} a_i' w_i \le W \\
\sum\limits_{i=1}^{I} a_i' \le A' \\
0.5 a_i \le a_i' \le 1.5 a_i
\end{cases}
\tag{6}
$$

where $f$ is the economic benefit, $max\,f$ is the maximum economic benefit, $i$ is the species of each crop, $a_i'$ is the optimized planting area of various crops, $w_i$ is the irrigation quota for each crop, $e_i$ is the purchase price per kilogram for each crop, $p_i$ is the yield per unit area for each crop (kg·hm$^{-2}$), $W$ is the volume of water available to the oasis, $A'$ is the farmland area within the carrying scale of the oasis, and $a_i$ is the planting area of each crop under the carrying area of the oasis. The above functions are programmed in the Python environment and call the *pulp* package to complete the calculation.

The carrying scales of the oases were selected by the $H_0^*$ (0.75), which are mentioned in Section 3.2.1, and is the lower limit for the oasis-carrying capacity level in a stable state. The oasis at this scale has depleted almost all the available unecological water and does not depend on groundwater exploitation. Under this situation, this carrying scale of 0.75 of the 5TOGs will be an ideal state representing a constraint for the extent of oasis expansion and will be considerably smaller than their actual scales.

As the farmland area and its proportion of each oasis and each crop for the 5TOGs has been quantified in the real scenario, and the carrying scales have been set up, the amount of farmland ($A'$) within the oasis-carrying scale can be identified by proportional scaling the farmland area. Hence, for the *pulp* function package to be able to obtain reasonable results, we define the swing altitude of $a_i'$ between 50–150% of $a_i$.

## 3. Results

### 3.1. Changes in the Area of 5TOGs and Their Farmlands in the Past 50 Years

#### 3.1.1. 5TOGs and Their Catchment Areas

Before assessing the change in the area of 5TOGs and their farmlands, we have evaluated the levels of available water resources of the 5TOGs from a watershed perspective as an oasis has to be supported by runoff. Just as the area of a watershed corresponds to its runoff output (Figure 2), an oasis's area should correspond to the runoff volume of the basin in which it is located. It is evident from Figure 3 that as the area of the oasis increases (x-axis), fluctuations occur in the corresponding catchment area and runoff. The area of HHO and TMNO is apparently large compared to the respective catchments' scale. Moreover, the area of 5TOGs in 2020 is negatively correlated with the area of the catchments (Figure 4) that support their existence (blue area). The ratios of the area of 5TGOs to their catchments are as follows: HTO (0.11:1), TMNO (1.12:1), HHO (0.43:1), AKO (0.30:1), and YKO (0.26:1). A similar trend was observed for the ratio of oasis area relative to runoff. It can be observed that the TMNO has the maximum expansion relative to the water resources of its basin.

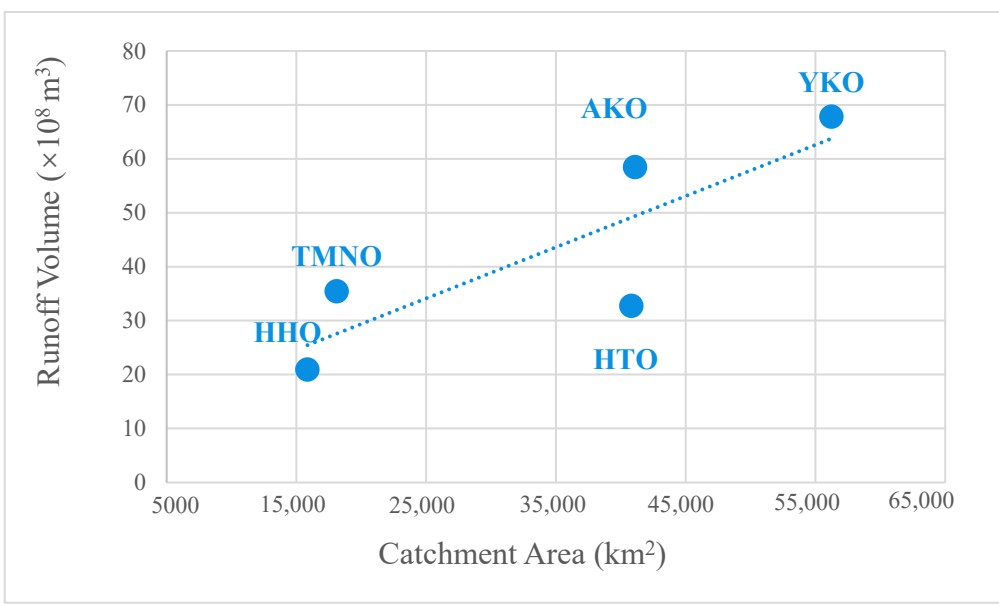

**Figure 2.** Scatter plot of runoff volume versus the catchment area for the 5TOG rivers.

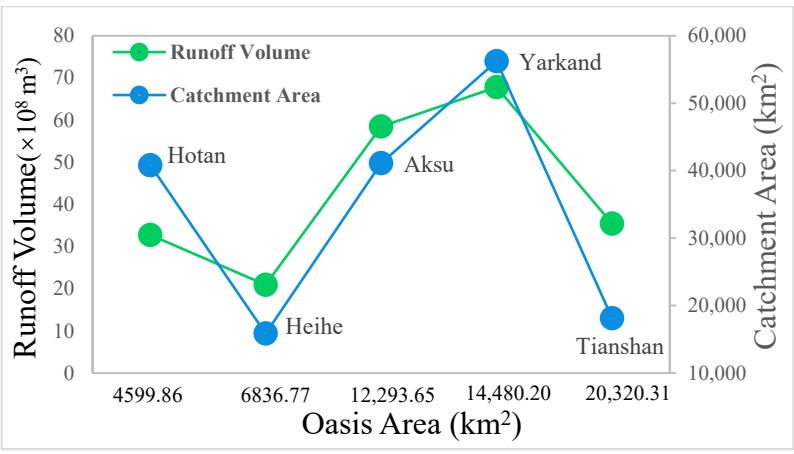

**Figure 3.** Biaxial scatter plot of 5TOG areas corresponding to the catchment area and runoff.

### 3.1.2. Changes in the Area of the 5TOGs and Their Oasis Farmland

Changes in the area of the 5TOGs and their farmland from 1980 to 2020 are presented in Figure 5a,b. It can be seen that the area of all 4TOGs increased considerably, except HTO. The most dramatic increase in the area occurred from 2010–2020. Although the area of HTO did not increase as compared to the other 4TOGs, farmland area in HTO increased, and it can be inferred that the land cover types within HTO are continuously transformed into farmlands. For AKO, YKO and TMNO, the increase in farmland area is consistent with the change in the oasis area. While the area of HHO has increased rapidly between 2010 and 2020, the area of its oasis farmland showed a steady increase from 1980 to 2020.

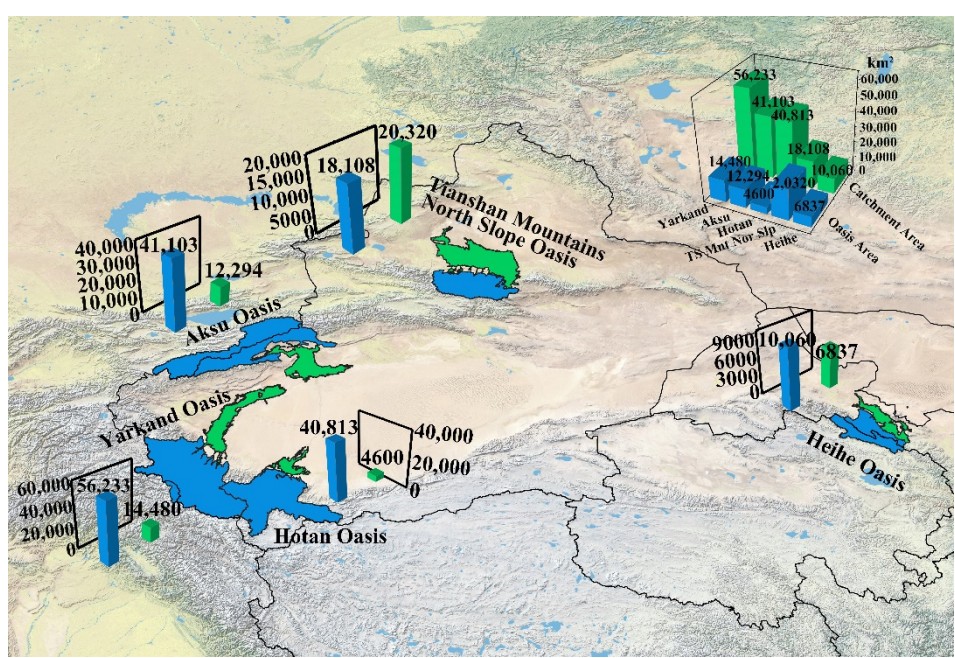

**Figure 4.** Comparison of the area of the 5TOGs with the area of their catchments.

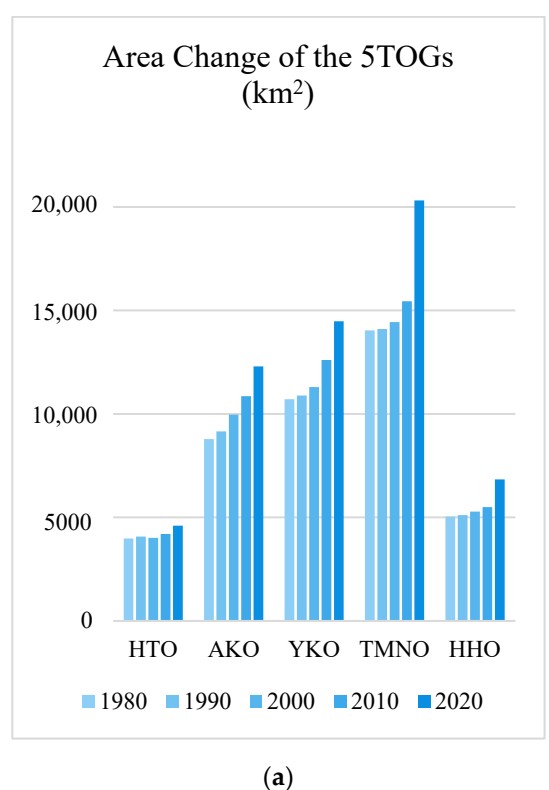

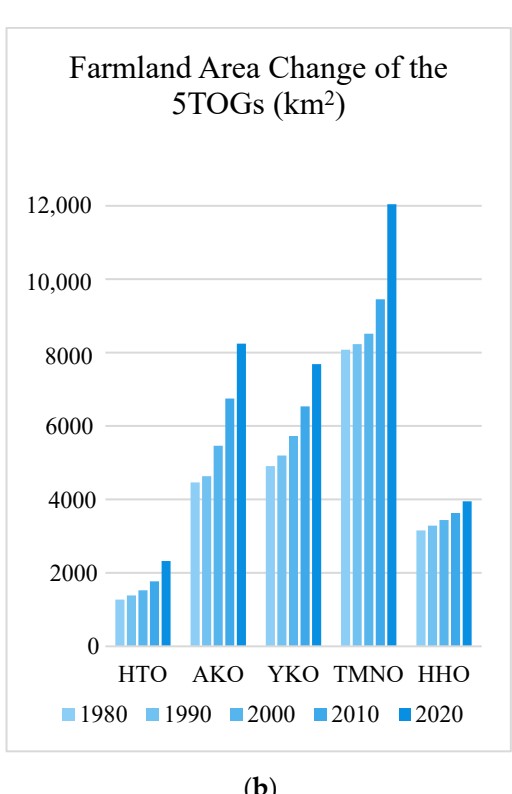

(**a**)

(**b**)

**Figure 5.** (**a**) Five changes in the area of the 5TOGs; (**b**) changes in the area of the 5TOGs' farmland.

### 3.1.3. Land Use/Cover Change (LUCC) Analysis of 5TOGs

To further investigate the expansion patterns of the 5TOGs, we have analyzed their LUCC using transfer matrices (Figure 6). Firstly, farmland remained the most important land use type in the 5TOGs from 1980 to 2020, and it has experienced a significant expansion with time. In 1980, the share of farmland in the 5TOGs was about 50%, and by the year 2020, the average value reached 70%. Secondly, a drastic reduction in the ecological land area occurred during this period, mainly converted to farmland with a more than 50%

conversion rate. Thirdly, the second primary source of agricultural land expansion is bare land, which represents the expansion process from oasis to desert areas.

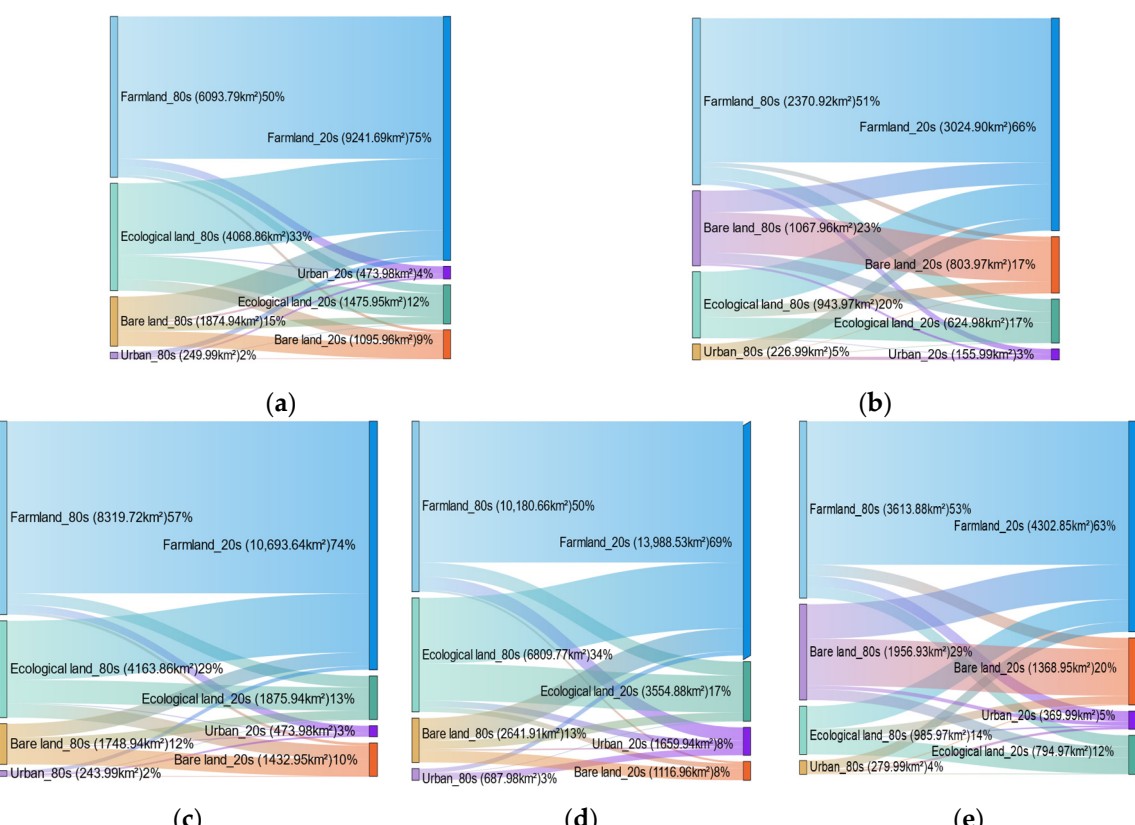

**Figure 6.** The transition of land use/cover O 5TOGs. (**a**) LUCC of AKO; (**b**) LUCC of HTO; (**c**) LUCC of YKO; (**d**) LUCC of TMNO; (**e**) LUCC of HHO.

*3.2. Evaluation of the Carrying Capacity and Its Suitable Scale in 5TOGs*

3.2.1. Carrying Capacity Dynamics of the 5TOGs

The carrying capacity of 5TOGs showed an overall decreasing trend (Figure 7). From 1980 to the 1990s, the carrying capacity of HTO and AKO was in a stable state, then turned into a sub-stable state in the 2000s, and in 2020 it was changed into an unstable state. The carrying capacity of YKO was in a sub-stable state from 1980 to 2010 and then entered into an unstable state from 2010 to 2020. It can be seen that HTO, YKO and AKO had maximum carrying capacity during the 1990s due to abundant runoff of the rivers in the Tarim Basin.

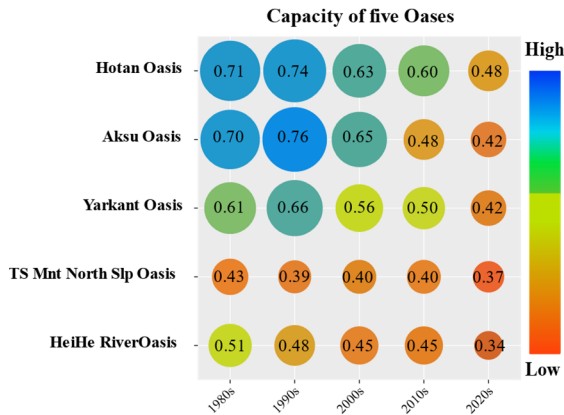

**Figure 7.** Oasis-carrying capacity of the 5TOGs from 1980 to 2020.

The carrying capacity of TMNO has been unstable since the 1980s, and its stability continued to decrease due to the increasing scale of the oasis. The carrying capacity of HHO was at the lower limit of the sub-stable level in the 1980s, then turned into a stable level in the 1990s. Although the carrying capacity of HHO was at a sub-stable level, it remained constant until the 2010s. However, the carrying capacity of HHO declined considerably during the 2010–2020 period and became the lowest among the 5TOGs.

### 3.2.2. Change in the Ground Water Storage of the 5TOGs

Due to the limitation of data availability, Figure 8 can only display the changes in groundwater storage in 5TOGs from the 2000s to the 2020s, which can be used to infer the changes in the carrying capacity of 5TOGs. The groundwater storage in AKO, TMNO and YKO experienced a substantial decrease from 2002–2020. The M-K test revealed significant variation in the groundwater storage changes in these three regions. The groundwater storage in TMNO showed a dramatic decline from May 2019. The groundwater storage in the HHO declined significantly, but the groundwater in the HTO area was not significant. Although groundwater storage changes in the above oases differ significantly, the overall trend is decreasing, indicating that the 5TOGs have exploited a large amount of groundwater to support agriculture.

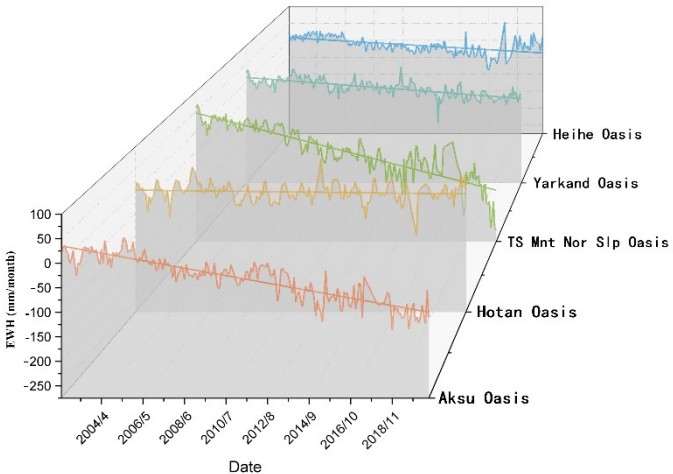

**Figure 8.** Change in Groundwater Storage of the 5TOGs from GREACE and GLDAS.

### 3.2.3. Suitable Scales of the 5TOGs

The threshold level of the suitable scales for the 5TOGs during each time period from 1980 to 2020, and the actual scale of the oasis at that time are shown in Figure 9. The actual scale of the 5TOGs is greater than their suitable scales since the 1980s. With time, the actual scales of the oases have exceeded their theoretical carrying scales while the suitable scales of the oases are decreasing. The actual scales of HTO and AKO slightly exceeded their appropriate scales between 1980 and 1990, after which the size of the oases increased substantially. The oasis scales of YKO, TMNO and HHO have significantly exceeded their suitable scales since the 1980s and have expanded since then. Among 5TOGs, TMNO is the largest in size and has exceeded its capacity to a large extent.

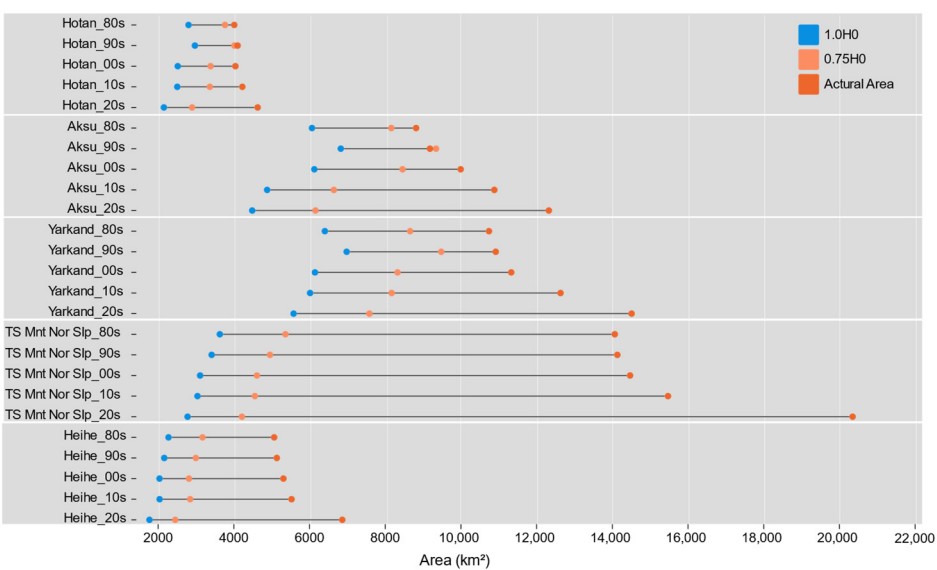

**Figure 9.** Suitable scale ranges and the actual scales of the 5TOGs.

### 3.3. Optimization of Planting Structure for 5TOGs under Their Suitable Scales

After quantifying the carrying capacity of the 5TOGs in northwest China and their suitable scales, we can propose optimized planting structures that maximize the economic returns of each oasis at the suitable scales through a linear programming algorithm based on the current planting structures of these oases.

#### 3.3.1. Current Planting Structures of the 5TOGs

The first step is to sort out the current agricultural cropping structure of the 5TOGs, including the major types of crops grown, acreage, crop yield, purchase unit price, yield per mu (666.67 m$^2$) and irrigation quota per mu information (Table A1). The AKO, HTO and YKO in the Tarim Basin grow cotton, jujube, walnut and wheat as main crops. Cotton, vegetables, maize, melons, and wheat are the main crops in TMNO. For HHO, the main crops are maize, vegetables, wheat, potatoes, herbs, barley, and oilseeds. Xinjiang is an important cotton-producing region in China, and the agriculture in the Hexi Corridor is dominated by grain cultivation.

#### 3.3.2. Optimized Planting Structures of the 5TOGs under the Suitable Scales

After analyzing the agricultural area and production statistics in 5TOGs, each oasis's most important crop species was obtained by weighting the crop factors of total yield, yield per unit area, and revenue and water consumption of each crop. The optimization of the planting structure based on the current planting area ratio of the crops was done for 5TOGS with a carrying scale of 0.75 to get an optimum planting structure scheme with maximum economic return (Figure 10).

In the AKO, YKO, and TMNO, the main crop should still be cotton (more than 60% planted), supplemented with apple, vegetables, and melon cultivation. In HTO, the cotton area should be reduced and replaced by walnut cultivation (34%), suitable for the local environment, followed by 31% corn cultivation. The HHO's cultivation structure is the most diverse, with a fair share of crops in the oasis agriculture, mainly food and vegetables, supplemented by cash crops (herbs and barley).

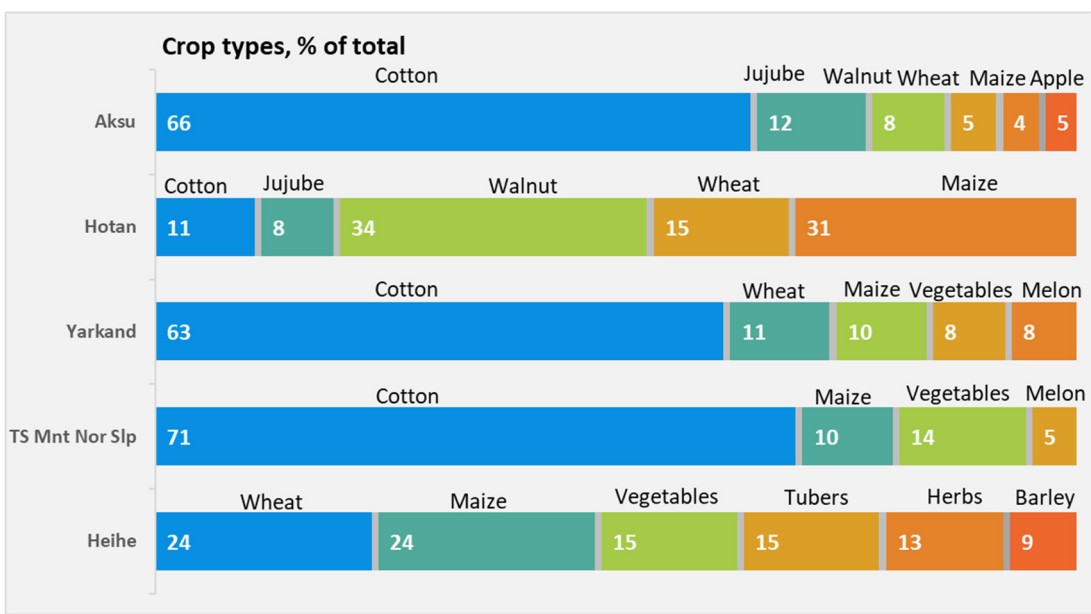

**Figure 10.** The optimal cropping pattern for the 5TOGs.

At the ideal suitable scales, the area of 5TOGs and its farmland are significantly reduced, thus meriting a significant reduction in oasis water consumption to suit local resource conditions. Optimizing the cropping scheme could reduce the water consumption of the oasis farmland by up to 50% of the available water resources with maximum agricultural yields. Consequently, the oasis would be able to maintain its ecological water security. (Table 3).

**Table 3.** Land and water resources pattern after scale restriction and planting structure optimization.

| | Farmland Area (km$^2$) | Oasis Area (km$^2$) | Agricultural Water Consumption ($\times 10^8$ m$^3$) | Residual Water Resources ($\times 10^8$ m$^3$) |
|---|---|---|---|---|
| Aksu | 4608.83 | 6127.81 | 22 | 29.45 |
| Hotan | 1882.6 | 2869.02 | 8.8 | 19.1 |
| Yarkand | 5579.17 | 7552.81 | 24 | 38.91 |
| Tianshan Mnt | 2880.6 | 4184.56 | 11 | 11.19 |
| Heihe | 1524.27 | 2421.89 | 6.8 | 12.63 |

## 4. Discussion

### 4.1. Optimization of Planting Structure for 5TOGs under Their Suitable Scales

The oases in Northwest China are unique landscapes providing human subsistence. Despite having limited water resources, oases in Northwest China have significantly expanded over the last 50 years. In this study, the runoff and catchment area corresponding to 5TOGs did not increase, but they showed fluctuation at HHO and TMNO data points. It indicates that these two oases (HHO and TMNO) have expanded beyond the limits of their water resources and are the two most intensively expanded of the 5TOGs. Our findings are consistent with Bie and Xie, and Zhang et al. [9,42]—they also reported an expansion of oases in HHO and TMNO.

The possible reason behind the TMNO expansion is the dense population and rapid economic development. It is considered the core area of industry, transportation, education, and science and technology in Xinjiangs' oases [43]. Furthermore, it is the key route connecting interior China, Central Asia, and European countries [44]. Due to the accumulation of population and availability of resources, the oasis has expanded dramatically to meet social development needs. Similarly, HHO is also a major agriculture commodity production base (e.g., grain, vegetable, melons, and oil crops) in China. It plays an important role in maintaining food security, thus guaranteeing socioeconomic stability in northwest China [45,46]. Being an artificial oasis suitable for human occupation and agriculture by human management [27], the expansion of HHO is inevitable.

Overall, the expansion of oases is mainly caused by the continuous development of cultivated land into woodland, which is evident from the expansion of the Gobi and other deserts [47,48].

Meanwhile, the land transformation trend of the 5TOGs showed a considerable shrinkage of ecological lands. This suggests that there has been a dramatic expansion in oases, with extensive tracts of farmland extending toward the oasis–desert ecotones [15]. Human intervention in the oasis landscape has become a serious concern for scientists due to its adverse impacts on the ecosystem [48,49].

It is worth noting that the town area of HTO declined from 1980–2020, which is contrary to what Maimaiti et al. [50] reported that the Hotan region had the largest town expansion rate and intensity. These two results seem inconsistent, but in fact, the land for farmers' residences in HTO was consolidated and thus transformed into urban land for the growth of farmland within the oasis. It can be seen that oasis agriculture has become an important industry in Northwest China.

### 4.2. Relationship between Results of Heat-Water Balance Model and GWSA of GREACE

The carrying capacity of 5TOGs was quantified using a water-heat balance model and based on results extracted from GRACE data. It was found that the period of decreasing groundwater storage (2000–2020) corresponded to the rapid decrease in the carrying capacity of the oases. Moreover, it was also found that the current scales of the 5TOGs have exceeded their suitable scales. Based on the above findings, it can be inferred that the overexpansion of the oasis caused the overloading of the carrying capacity of the 5TOGs. Furthermore, the dramatic increase in oasis farmland is the leading cause of the expansion of oasis areas [51]. To sustain that situation, groundwater extraction has played a vital role in the oasis farmland and overall oases expansion. Our temporal results demonstrate that groundwater pumping has substantially supported the expansion of 5TOGs since 2000, and this conclusion is consistent with the findings of Yang et al. [7]. In spatial analysis, the groundwater level in oases of Northwest China showed a declining trend in our study, which is consistent with the findings of Li et al. [52], and they reported a 5 to 35 m decline in the groundwater level of oases in northwest China from 2013 to 2019. Li and Luo [53] also reported that ecological security in Gansu in 2019 is considered to be under great ecological pressure, which is a red flag.

Oases farmlands rely on irrigated agriculture. Human technology (drilling wells, planting, mechanization, fertilizers, pesticides) consequently turned water resources (above and below ground) into agricultural output: enlargement of farming plots increased cultivated areas, decreased the number of grassland areas, hedgerow leveling, construction of trenches, etc. Reulier et al. [54] highlighted an obvious problem: the increasing agricultural production is achieved at the cost of ecological safety or the carrying capacity of the landscape, as water is the most important limiting factor supporting all this [28]. Reulier et al. [54] pointed out that ploughing up grassland into farmland increases the amount of potential permeation areas. However, such intensive agricultural interventions under limited water resources only weaken the carrying capacity or sustainability of the landscape rather than change the surface run-off process.

Additionally, Eigner and Nuppenau [55] pointed out that in the typical intensive agricultural area environment, there are few natural elements (such as forests or biotopes) left by humans to nature, and the narrow field margins only become the only seminatural habitats. Consequently, the expansion of oasis farmland may lead to the disappearance of ecotones and the homogenization of ecosystems across the landscape. It can be argued that the expansion of the 5TOGs was achieved at the expense of ecotone on the surface and groundwater storage on the subsurface.

The water-heat balance model used in this work is a simple and efficient method for assessing oasis carrying capacity. It captures the two most dominant limiting conditions for oases in arid environments: water scarcity and the surplus of heat. The existence of an oasis is a product of the equilibrium state of local water and energy conditions. In this artificial oases context, the area where water can flow determines the boundary to which oasis agriculture can be expanded [27]. Although this model has a simple structure, it is able to answer quantitatively the changes in the carrying capacity of the oasis. At the same time, the model requires data collection that meticulously distinguishes the non-ecological components of water resources to reasonably assess the state of the water-heat balance. When collecting data, the fact that oases often spatially span several administrative regions increases the difficulty of data collection but also increases the model's credibility. However, Wei et al. and Ariken et al. [56,57] point out that the bias toward large-scale statistics tends to limit studies to provincial administrative units and broad indicators, thus ignoring the complex local ecological reality. However, county-scale statistics are an efficient and cost-effective way to obtain spatially representative data in oasis landscape studies. It can be said that the water-heat balance model is an applicable method for rapid evaluation of the carrying capacity of oases.

### 4.3. The Implications of Optimizing the Planting Structure

In the past decades, human activities have significantly changed the distribution and allocation of limited water and land resources, which promoted economic development [58]. This is one of the important reasons why the expansion of oasis farmland has been able to proceed. In this study, we optimized the cropping structure for the maximum economic returns assuming that the oases can carry suitable scales. Although this scenario may seem idealistic, it is necessary. First, global agricultural water demand will increase over time as populations grow, incomes increase, and dietary preferences change. The competition between water demand for industrial and urban users and the ecological environment will be intensified [59]. Hence, we must achieve a water redundancy for future development to offset the negative effects of oasis expansion through human technology and productivity improvements.

Secondly, by limiting the oases' scales to their appropriate levels, we are paying back to nature, thereby safeguarding its ecological security. It is expected that a policy of returning farmland to the forest can be implemented to limit the size of cultivated land and reduce water use for agricultural irrigation [7,60]. According to some scholars, crop diversification can improve the provision of multiple ecosystem services that support resilience to abiotic stresses [61]. It is more important for oasis landscapes to maintain a fair share of ecological water in their water consumption process. Due to this, oasis scales ought to maintain a sufficient amount of water in the environment to improve the oases' carrying capacity.

The main cash crop in Xinjiang is cotton contributing 60% to China's cotton production [30]. It is the backbone of the socio-economic development in the region. This is why over the past 50 years, increasing population pressure has led to over-cultivation of the fragile desert–oasis transition zone for cotton cultivation [29]. At the same time, the expansion of cotton fields to alluvial plains is mainly influenced by farmers' choices, which are driven by crop price fluctuations, agricultural policies and natural conditions [62,63]. These factors are inevitable but must be considered to optimize planting structure. While maintaining planting structure optimization, we suggest that policies to control the moderate expansion of oases need to be strengthened, such as promoting the transfer of agricultural labor to non-farm labor and reasonably controlling food subsidies in these areas to effectively control oasis expansion [51]. By limiting the scales of oases and optimizing the cropping structure, the sustainable use of limited soil and water resources can be gradually enhanced and optimized, avoiding carrying capacity overload and maintaining ecosystem functioning [64].

## 5. Conclusions

This paper presents a quantitative analysis of the expansion and optimization of the planting structure of 5TOGs in Northwest China. In 5TOGs, the expansion of the oasis area from 1980–2020 was caused by the growth of farmland area. However, the most dramatic expansion of oases and their farmlands occurred during 2010–2020. The carrying capacity of 5TGOs exhibited a decreasing trend, while groundwater storage substantially declined in four oases except for HTO. The expansion of oases caused a significant decrease in the carrying capacity due to the overexploitation of resources, especially groundwater. Optimization of planting structure revealed that cotton should remain the main crop in AKO, YKO, and TMNO, with more than 60% area supplemented with fruit, walnut and vegetable cultivation. However, a reduction in the cotton cultivation area has been recommended for HTO. These findings provide a comprehensive analysis of oasis expansion which has practical significance for policy making regarding sustainable management of resources and development of agriculture in an oasis. Indeed, climate change, change in hydrological state, and human interference in the watershed considerably affect the oasis. Furthermore, the processes mentioned above also interfere with each other. However, in the present study, we have only focused on the changes occurring in the oasis state. Therefore, future research should investigate the groundwater cycle and soil water transfer response in arid zones regarding conceptual model uncertainty, input uncertainty, and parameter uniqueness.

**Author Contributions:** Conceptualization, B.L. and D.G.; methodology, B.L., D.X. and Y.L.; software, B.L., D.X. and Y.L.; validation, B.L., D.X. and Y.L.; formal analysis, B.L.; data curation, B.L., D.X. and Y.L.; writing—original draft preparation, B.L.; writing—review and editing, Z.A. and J.L.; visualization, B.L.; supervision, D.G. and J.L.; project administration, D.G.; funding acquisition, D.G. All authors have read and agreed to the published version of the manuscript.

**Funding:** This research was funded by the National Natural Science Foundation of China, grant number: 42171042.

**Data Availability Statement:** The data presented in this study are available upon request from the corresponding author.

**Conflicts of Interest:** The authors declare no conflict of interest.

# Appendix A

**Table A1.** Agricultural cropping structure of the 5TOGs.

| Aksu River Oasis Group | | | | | | Hotan River Oasis Group | | | | | | Yarkand River Oasis Group | | | | | | Tianshan Mountain Northern Slope Rivers Oasis Group | | | | | Heihe River Oasis Group | | | | |
|---|---|---|---|---|---|---|---|---|---|---|---|---|---|---|---|---|---|---|---|---|---|---|---|---|---|---|---|
| Crops | Area | Yield | Unit Price | Unit Yield | Irrigation Quota | Crops | Area | Yield | Unit Price | Unit Yield | Irrigation Quota | Crops | Area | Yield (ton) | Unit Price | Unit Yield | Irrigation Quota | Crops | Area | Yield | Unit Price | Unit Yield | Crops | Area | Yield | Unit Price | Unit Yield |
| - | (mu) | (ton) | (yuan/ton) | (kg/mu) | (ton/mu) | | (mu) | (ton) | (yuan/ton) | (kg/mu) | (ton/mu) | | (mu) | (ton) | (yuan/ton) | (kg/mu) | (ton/mu) | | (mu) | (ton) | (yuan/ton) | (kg/mu) | | (mu) | (ton) | (yuan/ton) | (kg/mu) |
| Cotton | 4,219,852 | 1,596,032 | 7500 | 378.22 | 430 | Wheat | 972,550 | 355,896 | 2300 | 439.13 | 270 | Cotton | 4,086,914 | 501,611.45 | 7500 | 421.33 | 305 | Cotton | 2,432,583 | 986,571 | 7500 | 426.05 | Maize | 1,541,800 | 788,558 | 1702.5 | 511.45 |
| Jujubes | 1,496,823 | 1,506,277 | 3274 | 1006.32 | 485 | Maize | 724,227 | 298,990 | 2100 | 495.41 | 255 | Wheat | 2,014,787 | 809,829.85 | 2407 | 462 | 245 | Vegetables | 266,125 | 1,258,759 | 1463.75 | 3511.99 | Vegetables | 414,800 | 1,292,670 | 1463.75 | 3116.37 |
| Walnuts | 1,546,525 | 327,471 | 9888 | 211.75 | 485 | Walnuts | 723,240 | 153,146 | 9888 | 211.75 | 365 | Maize | 1,862,267 | 943,324.6 | 2229 | 537 | 252 | Maize | 913,195 | 611,766 | 1702.5 | 669.92 | Wheat | 518,200 | 226,876 | 2182.5 | 437.82 |
| Wheat | 898,154 | 412,894 | 2183 | 459.71 | 375 | Jujubes | 529,845 | 83,385 | 4000 | 188.85 | 365 | Vegetables | 420,103 | 1,463,734 | 2824 | 3334 | 295 | Melonss | 98,090 | 40,2808 | 968.75 | 3504.78 | Tubers | 334,000 | 185,393.7 | 1050 | 555.07 |
| Maize | 676,565 | 421,273 | 1703 | 622.66 | 365 | Cotton | 238,859 | 21,934 | 7500 | 293.85 | 310 | Melonss | 438,854 | 1,264,976 | 2162 | 3344 | 245 | Grapes | 262,615 | 32,2918 | 1800 | 1229.63 | Herbs | 285,900 | 107,139.9 | 32,000 | 374.75 |
| Apples | 291,039 | 652,952 | 1838 | 2243.52 | 485 | Vegetables | 119,827 | 192,389 | 1500 | 1926.67 | 370 | Jujubes | 352,107 | 698,311.84 | 8334 | 211.75 | 440 | Wheat | 614,109 | 18,9781 | 2182.5 | 309 | Barley | 204,100 | 87,880.8 | 2300 | 430.58 |
| others | 330,451 | 521,203 | - | 1577.25 | 340 | Grapes | 113,925 | 127,794 | 2000 | 1346.09 | 365 | Walnuts | 22,519 | 454,561.34 | 9887.5 | 211.75 | 440 | Others | 631,426 | 91,324 | 1500 | 124.5 | Oil crops | 119,400 | 25,392.2 | 4766.667 | 212.66 |
| Vegetables | 272,552 | 675,319 | 1464 | 2477.76 | 420 | Melons | 51,040 | 97,456 | 969 | 2291.29 | 295 | Beans | 283,962 | 37,715 | - | 377 | 245 | Beets | 59,767 | 257,831 | 658.96 | 4317.26 | Grapes | 33,500 | 30,448.2 | 1800 | 908.9 |
| Pears | 189,916 | 469,742 | 2163 | 2473.42 | 485 | Apricots | 46,965 | 34,944 | 1000 | 892.85 | 515 | Tubers | 75,605 | 221,410 | - | 709 | 255 | Apples | 43,864 | 31,573 | 1837.5 | 719.79 | Beets | 21,400 | 51,341.3 | 658.96 | 2399.13 |
| Melons | 120,641 | 198,587 | 969 | 1646.1 | 340 | Oil crops | 37,401 | 3957 | 6412 | 126.96 | 270 | Apples | 71,149 | 192,346.88 | - | - | 440 | Tubers | 9222 | 37,671 | 1050 | 3977.09 | Pears | 28,400 | 11,386.8 | 2162.5 | 400.94 |
| Apricots | 77,273 | 163,279 | 1300 | 2113.01 | 485 | Apples | 14,250 | 10,787 | 1500 | 908.38 | 515 | Pears | 67,138 | 81,901.46 | - | - | 440 | Oil crops | 68,755 | 11,995 | 4766.67 | 174.47 | Jujubes | 17,600 | 9800.7 | 3274.125 | 556.86 |
| Beets | 30,750 | 129,668 | 659 | 4216.85 | 380 | Peaches | 8220 | 4116 | 7000 | 600.88 | 515 | Oil crops | 86,022 | 11,337 | - | 313 | 255 | Peaches | 14,835 | 23,592 | 1800 | 1590.29 | Apples | 4400 | 6818.5 | 1837.5 | 1549.66 |
| Oil crops | 52,951 | 8752 | 4767 | 165.28 | 360 | Pears | 1110 | 1152 | 1400 | 1245.41 | 515 | Peaches | 4278 | 32,518.64 | - | - | 440 | Matrimony vine | 18,300 | 3062 | 1800 | 163.75 | | | | | |
| Grapes | 28,772 | 64,009 | 1800 | 2224.7 | 445 | | | | | | | Grapes | 8953 | 21,368.67 | - | - | 340 | Jujube | 7481 | 1430 | 3274.13 | 191.15 | | | | | |
| Tubers | 18,031 | 34,207 | 1050 | 1897.12 | 360 | | | | | | | | | | | | | Beans | 4000 | 265 | 2500 | 66.16 | | | | | |

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
