# Peer review of "Analysis of the Expansion Characteristics and Carrying Capacity of Oasis Farmland in Northwestern China in Recent 40 Years"

_agronomy, doi:10.3390/agronomy12102448_

Round 1

Reviewer 1 Report

After revising the manuscript entitled "Analysis on the expansion characteristics and carrying capacity of oasis farmland in Northwestern China in recent 50 years" On overall, manuscript is very well written and the information provided is enough. The following minor changes are are suggested:

1. Change the ‘Agriculture’ to ‘Agronomy’ in the header of each page.

2. Bibliography page adjusted to vertical.

3. All references should be formatted according to journal requirements. In particular, the author's name should not be omitted.

4. Minor question: Could you include in the conclusion some idea about the importance of this work for considering UNCERTAINTY; specially under climate change scenarios?

Congratulations on your great work!

Reviewer 2 Report

Dear authors,   I reviewed your article with great interest. The topic you are investigating is very important and the outcomes can be of great impact on decision-making. In addition, although there is an excessive dependence solely on secondary data (some of which are of debateable reliability), there is plenty of good work to be appreciated. For instance, you laid out the case for their research, including a brief review of literature, very well. Also, the structure and style of the article are easy to follow and understand.   However, in my opinion, there are some areas in the article which requires necessary improvement. To begin with, the title promises a 50-year comparative analysis, while the earlies year the data covers is 1980. Year 1980 to 2020 would be 40 years, not 50 as mentioned in the title. Also, there several issues with the article's presentation, the figures in particular. Kindly find the remainder of my comments in a list below.   Major: - Section 2.3.3 (Linear programming for planting structure optimization): there is no sufficient explanation of the method. All is mentioned is the equation to maximise economic benefits, it is unclear to what constraints equation 5 is associated (equation 6 does not make sense) - Line 310: unclear to me what is the justification/explanation of why the authors are basing their optimized planting structures calculations on 0.75 carrying capacity while the existing numbers are way below that. - It is unclear how section 3.1.1 is useful to serve the objectives of the study. For example, there is no display of the changes over the years.   Minor: - Line 115/116 error in referencing of table 1. - Fig1: the map requires more contextual clarification, e.g. where is this region located within China. - Table1: data source of precipitation and potential ET is not described in section 2.2 (Data Sources) - Statement in line 174 requires citing. - Line 175: there is a point numbered as (2). What is point (1)? - Line 189: after the first coma, "maxf" needs to be added - Line 192-193: it is unclear what the variables W, A’ (capital letters), and ai (no apostrophe) are for? - Figure 3: the oasis names are missing. - Figure 4: map is unclear. - Figure 6: I do not think Sankey is the best diagram type to use for its purpose. - Figure 8: what are the numbers (unit) on the y-axis? - Figure 9: what are the numbers (unit) on the x-axis?   Best wishes

Round 2

Reviewer 2 Report

Thank you for considering the suggested comments.